# Home Parenteral and Enteral Nutrition

**DOI:** 10.3390/nu14132558

**Published:** 2022-06-21

**Authors:** Jamie Bering, John K. DiBaise

**Affiliations:** Division of Gastroenterology and Hepatology, Mayo Clinic, Scottsdale, AZ 85259, USA; bering.jamie@mayo.edu

**Keywords:** home parenteral nutrition, home enteral nutrition, enteral nutrition, parenteral nutrition, indications, complications, quality of life

## Abstract

While the history of nutrition support dates to the ancient world, modern home parenteral and enteral nutrition (HPEN) has been available since the 1960s. Home enteral nutrition is primarily for patients in whom there is a reduction in oral intake below the amount needed to maintain nutrition or hydration (i.e., oral failure), whereas home parenteral nutrition is used for patients when oral-enteral nutrition is temporarily or permanently impossible or absorption insufficient to maintain nutrition or hydration (i.e., intestinal failure). The development of home delivery of these therapies has revolutionized the field of clinical nutrition. The use of HPEN appears to be increasing on a global scale, and because of this, it is important for healthcare providers to understand all that HPEN entails to provide safe, efficacious, and cost-effective support to the HPEN patient. In this article, we provide a comprehensive review of the indications, patient requirements, monitoring, complications, and overall process of managing these therapies at home. Whereas some of the information in this article may be applicable to the pediatric patient, the focus is on the adult population.

## 1. Introduction

The evolution of home parenteral and enteral nutrition (HPEN) has been a critical medical advance for patients who are unable to maintain their nutrition by mouth. Technological advances during the 20th century have allowed these life-saving therapies to be delivered to patients at home, revolutionizing the field of clinical nutrition. Worldwide prevalence of home enteral nutrition (HEN) and parenteral nutrition (HPN) is difficult to estimate. In the United States, the most recent analysis from 2013 suggests approximately 25,011 patients receiving HPN and 437,882 patients receiving HEN [1]. The use of HPEN appears to be increasing on a global level [1,2,3,4]. Given the growing number of patients requiring HPEN, it is important for healthcare providers to understand what HPEN entails. In this narrative review, we will provide a comprehensive overview of HPEN, including indications, types of access, initiation of feeding, monitoring, complications, and socioeconomic considerations when managing these patients.

## 2. Indications and Requirements for HPEN

While consideration for home nutrition support is given to patients who are unable to meet their nutritional needs orally, that is, they have an appropriate indication for HPEN, other factors also must be considered, including insurance coverage, availability of laboratory and nursing support, and an acceptable home environment with satisfactory social support to ensure safe administration. Because HEN and HPN are complex and can be associated with complications, patient selection is important for a successful outcome. Patient assessment begins with a comprehensive nutritional assessment, determination of malnutrition/malnutrition risk, and identification of an indication for home therapy (Table 1) [5,6,7]. Some of the more common indications for HEN include severe oropharyngeal dysphagia due to primary neurologic disorders (vascular or degenerative) and malignancies (head and neck or esophagogastric), gastric outlet obstruction, and severe gastroparesis [8]. In contrast, parenteral nutrition is indicated in disorders where enteral nutrition is poorly tolerated or deemed inappropriate such as severe malabsorptive disorders including short bowel syndrome, high-output stoma or enterocutaneous fistula, severe intestinal dysmotility syndromes including chronic intestinal pseudoobstruction, and chronic bowel obstruction [9].

For the successful home delivery of nutrition support, it is critical that patients be capable of following the procedures for nutrition administration or have a caregiver who can perform these tasks, which are discussed in detail later [10]. As home nutrition support requires the home delivery of supplies, patients/caregivers need to be able to arrange for receipt of any needed formula and equipment. Because of the risk of complications with HPN, formal individualized training for the patient, caregiver, and/or home nurses has been recommended [6]. Patients receiving HEN should also undergo training provided by a multidisciplinary team [5]. Patients and/or their caregivers need to demonstrate an ability to properly administer their nutrition support therapy prior to hospital discharge [11,12].

Finally, HPEN is typically considered for patients who demonstrate a “long and indefinite” need in order for insurance coverage to be obtained [2,8,9]. More information regarding insurance reimbursement for home nutrition support within the United States is discussed below. Given the complexity of HPEN management, a multidisciplinary team is recommended to care for these patients [13].

## 3. Considerations When Initiating HPEN

### 3.1. Central Venous Access for HPN

For patients who require HPN, vascular access with a central venous catheter (CVC) is needed. Tunneled or subcutaneously implanted (i.e., ports) catheters with access to the internal jugular vein or subclavian vein are appropriate for long-term use. If a shorter duration of HPN is anticipated, typically <6–12 weeks, a peripherally inserted central venous catheter (PICC) is another option [14]. The use of a peripheral intravenous cannula or midline catheter is not appropriate for HPN as the tip of neither is appropriately situated in a central vein [14]. The use of femoral venous access is associated with a high risk of infection and venous thrombosis and is not recommended unless other access sites are not available [15]. While there is some controversy over the ideal location for the tip of central venous catheters, it is generally recommended that for PN, the catheter tip should terminate at the junction of the superior vena cava and right atrium to reduce the risk of thrombosis and dysrhythmias [10,14,16].

There are advantages and disadvantages to each type of CVC (Table 2). As such, patients should be included in the decision-making process. The proximal portion of tunneled CVCs exit the skin and, thus, have the injection cap external to the skin surface and may be less appealing from an aesthetic perspective but are generally preferred for administration of HPN. While ports may be appealing to some patients because they are subcutaneously implanted, these CVCs require a needle to be accessed and, in HPN, are generally accessed continually except during the once-weekly changing of the needle. Furthermore, the diaphragm membrane the needle punctures deteriorate over time and periodically may need to be replaced. In the case of a serious catheter-related bloodstream infection (CRBSI) where the CVC is unable to be salvaged, it also is more difficult to remove a port. The use of PICCs has increased over recent years related to their ease of placement and removal, accessibility, safety, and cost-effectiveness [17]. These catheters, however, have also been shown to have a higher risk of thrombosis compared to other CVCs, which limits their use for chronic therapy and are not recommended for HPN [18,19]. An important consideration when choosing a CVC is the number of infusion lumens present. The fewest number of lumens needed is recommended, preferably just a single lumen, as multilumen CVCs are associated with an increased incidence of CRBSIs [20].

### 3.2. Enteral Access for HEN

For HEN, the optimal route of administration depends upon the anticipated duration of use, adequacy of intestinal function, and risk of aspiration. Available routes of enteral access include transnasal and percutaneous. For patients who are likely to require 30 days or less of enteral nutrition, a nasoenteric tube is the preferred route for administration, although some individuals can be trained to insert and remove a nasogastric tube daily for nocturnal feedings. If HEN is anticipated to be needed >30 days, then a percutaneous route is generally preferred. Percutaneous gastrostomy (and sometimes jejunostomy) tubes can be placed surgically, endoscopically, or radiologically. Skin-level or low-profile gastrostomy devices (i.e., “buttons”) have become popular in individuals who require long-term enteral nutrition support.

Intragastric feeding, as opposed to feeding directly into the small intestine, is preferred, as it is more physiologic and can help activate the normal neural and hormonal pathways involved in digestion and absorption of nutrients [21,22]. Intragastric feeds also appear to buffer gastric acid more effectively and, thus, may provide ulcer prophylaxis [23]. Enteral access via the gastric route is generally easier to achieve compared with postpyloric tube placement. Postpyloric feeding is usually reserved for those who are intolerant of gastric feeding or those at high risk of aspiration of gastric contents.

### 3.3. Nutritional Formulation and Administration

An assessment of the patient’s calorie, protein, and fluid needs is needed prior to starting EN or PN. While indirect calorimetry is the gold standard for calculation of energy expenditure in critically ill patients, given its limited availability, it has become standard clinical practice for medically stable individuals to use predictive calculations (e.g., Harris–Benedict equation) or weight-based approaches [24]. Subsequent adjustments can be made to a patient’s formula based on weight trends, laboratory studies, urine output, and other clinical factors.

When initiating nutrition support in patients with severe protein energy malnutrition, both PN and EN need to be started slowly and cautiously, preferably in a hospital setting, to monitor for refeeding syndrome. Refeeding syndrome, discussed in more detail later, is a potentially fatal complication and consists of metabolic and electrolyte disturbances in response to the reintroduction of calories after a period of reduced or absent caloric intake [25]. While no standardized approach has been developed to guide the advancement of a nutrition regimen, several protocols exist [25,26,27]. Initiating PN at home requires a suitable patient (i.e., not at risk of refeeding and otherwise medically stable) and close monitoring by a willing patient, the PN ordering provider, and the home infusion company. If any of these is missing, the patient is best served by being hospitalized for initiating PN.

The ideal PN formulation should contain appropriate individualized calories as well as protein requirements and adequate volume while avoiding excessive dextrose and lipids. Trace elements and vitamins are also required in all PN solutions. Insulin and other additives are sometimes included. Typically, HPN is cycled/infused overnight, often over a period of 10 to 14 h via a programmable infusion pump. This will require an increase in the rate of PN infusion to decrease the infusion time, thus allowing the patient to have more independence from the infusion pump. Nocturnal HPN does have its drawbacks, however, including increased interruption of sleep due to increased urination and noise from the infusion pump. As such, some patients prefer to infuse PN during the day. Portable pumps that can be carried in a backpack or tote are available for those who need to infuse PN during the day. Of note, home infusion pumps are usually programmed to infuse the formula with a gradual infusion taper down over the final one or two hours. The taper-down period helps to avoid rebound hypoglycemia when the infusion is finished. In adults, a ramp-up mode is rarely necessary.

For patients receiving EN, several methods for formula administration can be employed. Perhaps the most physiologic technique is bolus administration. This involves using a syringe to infuse a 200–400 mL bolus of formula over 5–15 min at various times throughout the day, similar to how one would eat regular meals. Bolus feeding is preferred for intragastric feedings in active, alert patients with low aspiration risk. For patients receiving gastric feeds who are intolerant to bolus administration, gravity feeding is an alternative option allowing feeds to be administered more slowly into the stomach via gravity flow, generally over 30–45 min. Finally, an infusion pump can be used to administer EN at a controlled rate and is the preferred method for patients receiving jejunal feeds and those receiving gastric feeds who are intolerant to both the bolus and gravity methods. Notably, patients may experience gastrointestinal symptoms such as abdominal distension, cramping, and diarrhea if the rate of feeding is too high. Similar to HPN, most HEN consumers prefer cycled nocturnal feeding as it allows maximal use of the gut and allows normal activities during the day. Rebound hypoglycemia is uncommonly seen in the HEN setting. Compact portable infusion pumps that can be carried around in a backpack or large tote are available for the patient who requires continuous infusion or prefers to infuse during the day.

### 3.4. Patient and Caregiver Training

As noted previously, training of the patient and their caregiver is an essential part of providing HPEN as these are complex nutritional therapies with the potential for serious complications. It has been shown that many potential complications of both HEN and HPN can be minimized or prevented entirely by providing a comprehensive educational curriculum to patients [28,29,30,31]. As part of this process, a psychosocial assessment of the patient’s and caregiver’s physical and mental capacity and ability to administer HPEN is important. Training should encompass topics such as hand hygiene, care of CVCs and/or enteral access devices and sites, formula administration, complication monitoring, and who to contact with questions or concerns [28]. Table 3 includes general education topics that should be covered for consumers who will be administering HPN.

A typical training program should be incorporated into the decision making and discharge planning for all HPEN patients. Training may occur during hospitalization or after the patient has been discharged home by their infusion company and home nurse depending on the institution and available resources. Wherever the education takes place, periodic assessment of patient and/or caregiver knowledge of important elements of HPEN delivery and monitoring should be performed. Several effective teaching strategies have been identified that can be incorporated into the caregiver education process [32]. Patients and/or their caregivers should be required to redemonstrate proper technique and gain approval from their nurse educator(s) before they are allowed to administer HPEN independently at home. If patients are unable or unwilling to perform the necessary tasks for safe administration of home nutrition support, other options should be explored.

## 4. Transitioning from Hospital to Home

In the patient determined to go home on EN or PN, once the patient has demonstrated tolerability to the optimized formula that has been cycled, arrangements can begin for their transition to home. Several factors need to be considered when transitioning a patient from an acute care setting to home. A safety assessment of the home environment is necessary to ensure that patients have the proper resources to continue EN/PN administration after discharge [13]. For U.S. consumers, insurance coverage should be determined as soon as possible to prevent unnecessary hospital discharge delays. This often requires the involvement of a case manager or social worker. A homecare agency and, for patients requiring PN, a home infusion company will need to be selected to assist with formula and supply delivery. As mentioned previously, formal patient and caregiver training should be employed including written and verbal instructions to follow and when to contact their outpatient nutrition support team for questions or concerns [13]. An important step before hospital discharge is to make sure the patient has follow-up care arranged with a physician who will also take responsibility for monitoring laboratory values and the patient’s clinical status and adjusting the PN/EN formula as needed. This physician needs to work closely with a pharmacist, dietitian, and home care nurse when the patient finally goes home.

## 5. Monitoring HPEN

### 5.1. HPN Monitoring

Patients receiving HPN require routine monitoring that should include body weight, urine output, biochemistry lab studies including micronutrient levels, and bone density measurement [6]. Various monitoring approaches have been suggested; however, none are evidence-based and generally are institution-specific [5,6,26,33]. From a routine laboratory monitoring standpoint, we suggest weekly monitoring of electrolytes and liver enzymes initially following hospital discharge and as the patient’s condition stabilizes, usually after 1 month, the monitoring interval can be increased to every other week, for another 1 to 2 months, then monthly thereafter for most patients. In some, a quarterly schedule is acceptable [28] (Table 4). A complete blood count is usually checked quarterly. Micronutrient levels (vitamins A, B12, folate, D and E, and iron, zinc, copper, and selenium), essential fatty acids, and triglyceride level are checked at baseline and at least annually; more often if deficiencies are identified and supplementation ongoing [33,34,35] (Table 4).

With respect to blood glucose monitoring, levels should also be monitored in patients on HPN to check for hyperglycemia as well as rebound hypoglycemia. Patients should be instructed to check their blood glucose about 1 h after starting the PN infusion and about 1 h after stopping the infusion in all patients who are receiving cyclical PN. If glucose levels remain stable after 1 to 2 weeks, then patients can discontinue monitoring unless a change in PN formula is made. When blood glucose levels are consistently above 180 to 200 mg/dL, regular insulin is often added to the PN formula; the amount should be individualized. Importantly, patients who receive insulin with their PN infusions should continue to monitor their blood glucose levels routinely with each infusion as adjustments may be needed [10].

Patients on long-term PN also require monitoring for the development of metabolic bone disease, including osteoporosis and osteomalacia [34]. One recent study of patients with intestinal insufficiency and intestinal failure requiring PN described a nearly 57% prevalence of osteoporosis [36]. Risk factors include inactivity, malabsorption, low body mass index, chronic inflammation, medications (e.g., corticosteroids), low vitamin D levels, and altered calcium homeostasis, among others [34]. Screening for metabolic bone disease is most commonly performed using bone density testing by dual-energy x-ray absorptiometry (DXA) of the hip and lumbar spine [37]. Guidelines exist that provide recommendations on use of DXA to define osteoporosis, to periodically monitor bone density, and to identify thresholds at which to start specific osteoporosis treatments [38].

### 5.2. HEN Monitoring

Like patients receiving HPN, patients receiving HEN should also be monitored regularly by a multidisciplinary team. Body weight and hydration status should be routinely evaluated to determine adequacy of the EN [5]. Adjustments in enteral formula and water administration can be tailored to each patient. Similar to patients receiving long-term PN, monitoring of routine laboratory studies and micronutrient levels in all patients receiving HEN should be obtained at least annually with more frequent monitoring and supplementation when deficiencies are identified [10]. Bone density screening should also be pursued at regular intervals.

## 6. Complications

Although HPEN can be life-saving therapies, they are also associated with complications that can be life threatening. An awareness of these complications and close monitoring is essential for prevention and identification of these problems. Here we provide a summary of some of the more common complications.

### 6.1. Central Venous Catheter-Related Complications

Complications of long-term central venous catheters (CVCs) include infection, thrombosis, and malfunction/breakage/dislodgement. One of the most common reasons for hospitalization in patients on HPN is catheter-related bloodstream infection (CRBSI) [39]. According to the most recent International Nosocomial Infection Control consortium surveillance data, there are approximately 4.1 central line-associated bloodstream infections per 1000 central line days [40]. Infection-related complications can be reduced with proper catheter insertion and care, as contamination of the catheter hub and migration of skin organisms into the catheter tract are the major events that lead to bacteremia [41]. Other sources of infection include contamination from infused substances or hematogenous spread from an unrelated infectious source [42]. The type of CVC and certain other catheter-related factors, including larger catheter caliber, absence of a cuff, and increased number of lumens, also correlate with increased infection risk [41]. Typical symptoms that suggest infection include fever, malaise, and tachycardia during CVC use, although these may not be present in all cases of CRBSI [43]. Elevation in leukocyte count and liver enzymes is often seen. Blood cultures should be obtained both from the CVC and from a peripheral vein prior to initiating empiric antibiotic therapy [44]. The most common causes of CRBSI include coagulase-negative staphylococci, *Staphylococcus aureus*, and *Staphylococcus epidermidis*. Gram-negative and fungal infections are less common.

If CRBSI is confirmed, the antibiotic therapy should be tailored to the identified pathogen. A 14-day course of intravenous antibiotic therapy is recommended in most uncomplicated cases. It has been recommended that PN be temporarily discontinued in the setting of a newly diagnosed CRBSI; the optimal number of days of discontinuation remains uncertain. Although CVC removal and replacement may be required as part of the treatment plan, repeated line removal and replacement procedures can threaten vascular access over time [45]. As such, for those patients on HPN and/or those who have existing vascular access challenges, an attempt at catheter salvage should be entertained. Catheter salvage success is primarily dependent on the culprit organism and therapeutic susceptibilities [43]. In a recent review and metaanalysis evaluating CVC salvage in adult patients on HPN, it was found that successful catheter salvage rates were highest in patients with coagulase-negative staphylococci followed by Gram-negative bacteria, while the lowest rates of successful catheter salvage were seen in patients with *S. aureus* and polymicrobial infections [46]. One technique for CVC salvage, as well as CRBSI prevention, has been the use of antibiotic lock therapy [43,47,48]. Attempted catheter salvage is not recommended for critically ill, unstable patients or those with fungal line infections [49]. Use of ethanol lock as a preventative and salvage agent has gained recent interest as an alternative to an antibiotic lock [50]. The decision to attempt line salvage may best be made with input from an Infectious Diseases specialist.

In addition to CRBSI, infection of the catheter tunnel and exit site can occur. Infections at the exit site generally present with purulent drainage, while tunnel infections manifest with erythema and tenderness along the catheter tunnel tract [51]. If drainage is present, culture of the site can be obtained to help target antimicrobial therapy. Exit-site infections often resolve with systemic antibiotics, and line removal can often be avoided. In contrast, infections of the catheter tunnel necessitate removal of the line in addition to systemic antibiotic treatment [51].

Deep venous thrombosis (DVT) is another common complication of indwelling CVCs. The reported incidence of this complication is variable due to the heterogeneity of the literature. Some studies suggest an incidence as high as 67%, while more recent studies report a rate of 14–18% [52,53]. Symptomatic DVT occurs less often, 1–5% of patients with a CVC, and may present with a variety of symptoms, including phlebitis, swelling, and catheter malfunction, depending on the severity of the thrombosis [54,55]. Risk factors for thrombus formation include catheter type, insertion site, tip location, and underlying patient factors such as the presence of a hypercoagulable state (e.g., malignancy, sepsis, critical illness, thrombophilia), prior venous thromboembolism, and concomitant use of certain medications such as oral contraceptives or hormone replacement therapies [53,56]. Several studies have shown that PICCs carry a higher risk for thrombosis compared to tunneled and subcutaneously implanted CVCs [18,57,58]. Other catheter-related factors that increase the risk of thrombosis include larger catheter diameter, tip location proximal to the superior vena cava, and access site, with femoral location being the highest risk [53]. Duplex ultrasonography typically is the diagnostic test of choice; however, venography is an option when suspicion remains high despite a negative ultrasound evaluation. Anticoagulation is typically recommended for the treatment of DVT when found; however, the duration may vary depending on other clinical circumstances [59,60,61]. Several guidelines suggest that CVCs can remain in place despite thrombus formation if the line is still needed, functional, well positioned, not infected, and symptom resolution occurs (if present) [59,62].

Damage to or deterioration of CVCs may occur over time due to normal wear and tear with prolonged use, repeated clamping, or use of high-pressure flushing in occluded catheters [63]. Catheter fracture may be minor, involving only the external sheath, or can involve full rupture of the catheter. Because repeated CVC loss can compromise vascular access over time, catheter salvage, when possible, should be considered. For catheters with minor damage involving the external sheath, catheter repair kits are available. This procedure can be done in the office by a trained provider. Data are emerging that repair is safe and can prolong the longevity of the CVC [63,64].

### 6.2. HEN Complications

Patients receiving HEN may develop complications related to the tube or tube feeding. Although some complications are universal for each tube type and route of access (e.g., nasoenteral, percutaneous), including the risk of clogging and accidental displacement, each type of access also carries its own unique complications [65]. In the HEN scenario, placement of a percutaneous tube, either gastrostomy or jejunostomy, is more suitable than nasoenteral placement for obvious aesthetic and comfort reasons.

Percutaneous tube complications can result from their placement or subsequent long-term use. While the majority of complications related to percutaneous tube placement are minor, major complications may occur in up to 4% of procedures [66]. Potential problems during and after percutaneous placement include perforation/injury to surrounding organs, wound infection, hemorrhage, leakage at the tube site, and buried bumper syndrome.

Infection is one of the most common complications of percutaneous feeding tubes, ranging from 12 to 32% in gastrostomy tubes [67]. Infection can occur at any time after placement. Most infections are minor and typically present with symptoms of tenderness, erythema, and purulent drainage at the stoma site. Sometimes, however, it can be difficult to discern infection from other non-infectious causes of peristomal irritation. To help address this diagnostic dilemma, Mundi et al. have proposed objective criteria to help systematically evaluate patients for peristomal infections [68]. These criteria include an assessment of erythema, induration, and exudate at the stoma site, culminating in a score for risk stratification of the patient. Culture of the peristomal drainage is generally not advised as it typically includes skin contaminants. Treatment of peristomal infections with antibiotics, administered either orally or via the tube, is usually successful and hospitalization or use of intravenous antibiotics is uncommon. The tube rarely needs to be removed to facilitate infection control. Risk factors that predispose to the development of peristomal infection include underlying immune suppression and increased pressure from the bolster or traction on the tube [69]. Studies have shown that antibiotic prophylaxis given at the time of initial tube placement can reduce the incidence of peristomal infections [70].

Bleeding complications have been reported in up to 3% of percutaneous tube placement procedures [71]. Immediate postprocedural bleeding can result from injury to an abdominal wall vessel or originate within the tract itself, while delayed bleeding is usually related to gastric mucosal ulceration from a tight internal bumper. Other rare bleeding complications from percutaneous gastrostomy tubes that have been reported include rectus sheath hematoma, esophageal ulceration, intraabdominal vascular injury, and abdominal wall pseudoaneurysm formation [72,73,74,75,76]. Bleeding may manifest as oozing at the tube site or through the tube itself, hematemesis, melena, anemia, or hemodynamic instability [69]. Depending on the severity of bleeding, management may include applying external pressure or temporarily tightening the external bolster against the skin for a tamponade effect for peristomal oozing, or may require endoscopic, radiologic, or surgical intervention. For patients who use chronic antiplatelet and/or anticoagulant agents, guidelines put forth by the American Society of Gastrointestinal Endoscopy advise on length of medication held prior to tube placement [77].

Inadvertent feeding tube removal occurs in up to 12.8% of patients and can result from accidental traction applied to the tube or, in the case when an internal balloon bolster is present, if the balloon breaks and is no longer able to secure the tube in place [78]. If dislodgement of the tube occurs prior to tract maturation, which can take approximately 4 weeks, blind replacement should not be attempted due to the high risk of a misplaced tube as the stomach and abdominal wall may have separated. Patients should be treated with antibiotic therapy and monitored for peritonitis. If symptoms of peritonitis develop, surgical consultation should be requested. In asymptomatic patients, a new tube can be placed after a few days of monitoring. Immediate replacement, either endoscopically or radiologically, after inadvertent removal has also been reported but should be decided based on the individual clinical circumstances [79,80].

Percutaneous tube malposition and injury to internal organs can occur during tube placement. Reported injuries have involved the colon, small bowel, liver, and spleen [81]. Diagnosis can usually be achieved using computed tomography or fluoroscopic imaging with oral contrast or contrast injected through the tube [82]. In addition to malposition, in patients who have gastrojejunal tubes, the distal tip of the jejunal extension may displace back into the stomach. This frustrating, often recurrent, problem can be managed by tube repositioning and clipping into place or removal and replacement with a jejunostomy tube. Finally, buried bumper syndrome is an uncommon complication of percutaneous feeding tubes where the internal tube fixation device migrates into or through the abdominal wall. The estimated incidence of this complication is between 0.3 and 2.4% and often is related to increased tension between the external and internal bolsters [83]. Presenting symptoms can include pain with inability to manipulate the tube, leakage around the tube, and loss of tube patency and complications including bleeding, abscess, peritonitis, and perforation may occur. Endoscopic evaluation is sometimes needed to determine the depth of tube migration and most appropriate treatment option [83].

In addition to tube complications, it is also important to consider the potential gastrointestinal symptoms that can occur because of EN. Patients may develop bloating, diarrhea, constipation, flatulence, nausea, vomiting, and reflux. Some patients experience aspiration of tube feeding. These symptoms can limit a patient’s ability to tolerate EN and lead to undernourishment. The CAFANE study sought to evaluate variables that would impact the risk of these adverse events for patients receiving HEN and determined that type of enteral formula and route of administration were important contributing factors [84]. A variety of strategies may be helpful in mitigating these GI symptoms and include adjusting the infusion rate of enteral feeds, changing the enteral formula, and using medications such as antacids or antidiarrheals where appropriate.

Finally, inappropriate handling of the formula may lead to contamination or spoilage [10]. In order to prevent this from occurring, patients should be educated on appropriate formula handling and storage. Using effective hand hygiene during EN preparation and administration is crucial. Not only has a correlation been shown between healthcare staff hand cultures and EN contamination in the hospital setting, but there was also a reduction in contamination for patients who received an infection control program [85]. Other important strategies include using a clean workspace for EN preparation and using equipment dedicated only for EN use. Storing the EN formula according to the manufacturer’s instruction, following recommended hang times when using an open system, and avoiding “topping off” remaining formula can also reduce complications related to formula contamination or spoilage [13].

### 6.3. Intestinal-Failure-Associated Liver Disease

Intestinal-failure-associated liver disease (IFALD) describes patients with chronic intestinal failure, usually receiving long-term PN, who develop liver dysfunction. Patterns of liver injury include steatosis, cholestasis, or a combination of these [86]. Cholelithiasis and gallbladder sludge are also common. The diagnosis should be based on the presence of abnormal liver biochemical tests and/or evidence of radiological and/or histological liver abnormalities occurring in an individual with intestinal failure, in the absence of another cause [87]. Risk factors for IFALD include overfeeding, type and amount of intravenous lipid emulsion administered, duration of PN use, lack of enteral stimulation, and recurrent sepsis [86]. Optimizing the PN formula, using non-soy-based lipid emulsions and limiting the amount of intravenous lipid to <1 g/kg/day, encouraging oral intake/enteral stimulation, normalizing micronutrients, and cycling the delivery of PN are all recommended to prevent and treat IFALD [88]. For those with refractory liver injury in the setting of intestinal failure, combined intestine-liver transplantation may be needed.

### 6.4. Metabolic Bone Disease

As mentioned previously, patients receiving chronic PN and EN are at risk of developing metabolic bone diseases including osteomalacia, osteopenia, and osteoporosis. Bone loss in patients on chronic HPN may be related to the nutrient composition of the PN with an often-negative calcium balance and vitamin D deficiency [89,90]. Other factors that may contribute or predispose patients to this condition include previously existing metabolic bone disease, malabsorption, medications such as corticosteroid use, and hypercalciuria [91,92]. Regular screening with DXA can help identify patients with metabolic bone disease and determine when treatment with specific osteoporosis therapy is appropriate [38].

### 6.5. Fluid and Electrolyte Derangements

Fluid disturbance can present as either hypovolemia/dehydration or hypervolemia. Patients at risk for developing dehydration are those with increased GI losses, intestinal malabsorption, fluid restrictions, advanced age, or altered mental status [89,93]. For those who have underlying cardiac, liver, or renal disease, hypervolemia may be of more concern [94]. It is important for patients and their caregivers to be educated on monitoring fluid intake and output and signs and symptoms of dehydration and fluid overload. Nutrition support can be adjusted to help optimize the fluid balance by changing fluid volume administered, adjusting the formula and/or electrolyte content administered, and using medications and/or supplemental oral rehydration solution or parenteral fluids when appropriate [89].

Electrolyte derangements, particularly involving sodium, potassium, magnesium, and phosphorus, are common in the setting of HPEN, especially in the early stages of therapy. Electrolytes are involved in several essential bodily functions and when derangements are severe, there can be significant clinical consequences including fatality [95]. Several factors can contribute to electrolyte abnormalities such as renal function, underlying disease process, changes in disease acuity, and medications [89]. A comprehensive understanding of these complex processes is required to effectively manage electrolytes. As noted previously, routine laboratory monitoring allows identification and correction of any disturbances that might occur. This can be done more frequently when first starting EN or PN or when making changes to the formula. Once stabilization of electrolyte levels has been established, the monitoring interval can be spaced out. Suggested laboratory studies to follow are included in Table 4.

### 6.6. Vitamin and Trace Element Disturbances

Micronutrient deficiencies are encountered regularly in patients receiving HPEN; however, they are often clinically silent. Thus, a high index of clinical suspicion and periodic laboratory monitoring is needed to identify, as mentioned previously. Since the introduction of home PN infusions, commercially available lipid, multivitamin, and trace element preparations have undergone substantial modifications in order to provide the essential nutrients and reduce the potential for micronutrient deficiencies. Consequently, micronutrient deficiencies described in the early years of PN such as iron, selenium, copper, zinc, thiamine, copper, vitamin A, vitamin E, vitamin D, and essential fatty acid are now less common. Importantly, most commercial trace element solutions only provide the recommended daily nutrient need without consideration of replacement of deficient micronutrient stores, and additional supplementation may be necessary. In addition, when HPN patients are completely or incompletely weaned from PN, they are at risk for developing deficiencies as determined by their underlying bowel condition and clinical status, and life-long oral supplementation and monitoring of micronutrient levels are necessary (Table 4). This is particularly important considering recent shortages in parenteral individual and MTE preparations available for use in PN.

Manganese toxicity is a rare complication that can occur with long-term HPN use and can lead to accumulation in various organs over time such as the liver, brain, and bone [7]. Neurotoxicity from hypermanganesemia is well documented with patients exhibiting symptoms including motor function deficits, mood destabilization, and cognitive impairment. The recent availability of a multiple trace element product for PN that does not contain manganese may lower the risk of this complication, but periodic monitoring is still recommended.

### 6.7. Refeeding Syndrome

As alluded to previously, refeeding syndrome can be defined as metabolic and electrolyte alterations occurring because of the reintroduction and/or increased provision of calories after a period of decreased or absent caloric intake [25]. Refeeding syndrome can be life threatening and is a risk for patients receiving both PN and EN. Because there is no standardized definition, the true incidence is uncertain [96]. Hypophosphatemia, hypokalemia, and hypomagnesemia are the classical electrolyte disturbances that occur in refeeding syndrome, although patients may also develop thiamine deficiency and alterations in fluid, glucose, protein, and fat metabolism [97]. Several risk factors for the development of refeeding syndrome have been identified and include low body mass index, unintentional weight loss of at least 10–15% body weight, little or no nutritional intake, and low levels of potassium, phosphate, or magnesium prior to initiation of nutrition support (Table 5) [26]. Recommendations on how to reduce the risk of refeeding syndrome include gradually initiating and advancing the nutrition support along with adequate monitoring and replacement of electrolytes [26,27,98,99]. Importantly, patients at risk of refeeding syndrome should also receive thiamine supplementation, and electrolyte deficiencies should be repleted both prior to and after initiation of feeding. If refeeding syndrome does occur, nutrition and fluid administration should be reduced, and metabolic derangements addressed prior to advancing the formula [100].

## 7. HPEN Cost

The financial cost of HPEN is a particular concern for the U.S. consumer. For those receiving Medicare, specific criteria must be fulfilled for insurance coverage, and reimbursement to be obtained [101]. In September 2021, the Durable Medical Equipment Medicare Advisory Contractors released new local coverage determinations (LCD) for both HPN and HEN [8,9]. Medical documentation must clearly address the indication for nutrition support, its anticipated length of need, and that such support is both “reasonable and necessary” [8,9,101]. While the cost of EN can be comparable to the cost of a typical monthly grocery bill, PN is often cost-prohibitive without insurance assistance. Previously, Medicare coverage required that the HPN or HEN be the main source of nutrition for at least 3 months; however, under the recently updated LCD, patients must meet the “test of permanence” with an underlying “permanent” disorder that will require “long and indefinite” nutrition support [8,9,101]. Importantly, requirements change periodically, and it is best for an appointed team member to keep the updated insurance coverage parameters available to ensure proper documentation and support for the patient’s needs.

## 8. HPEN Quality of Life and Survival

Several generic and disease-specific instruments and questionnaires exist to help measure and monitor a patient’s quality of life in relation to various illnesses and health status. The literature is inconsistent regarding the quality of life of HPEN patients, perhaps in part due to the different instruments used to assess this parameter [102]. For those patients who report a poor quality of life compared to the general population, the underlying disease process and complications have been identified as major contributors, in addition to the HPEN itself [103]. A recent systematic review evaluated the effect of HEN on the health-related quality of life and found that EN was associated with an improvement in quality of life [104]. Factors specifically related to HPEN that seem to affect the quality of life include infusion time of PN/EN formula and duration of dependence on EN/PN. Other factors that may affect a patient’s quality of life are included in Table 6. Ongoing support of patients and their caregivers both at the time of HPEN initiation and longitudinally thereafter can help to reduce the negative impact that these therapies may have on patients’ everyday lives [105]. Several organizations, such as the Oley Foundation (www.oley.org), provide educational resources and support. Involvement in such groups has been shown to be associated with an improvement in the quality of life of the HPEN patient [106].

Because HPEN can have a significant impact on the physical, social, and psychological aspects of a patient’s life, it has been recommended that quality of life be assessed at the beginning of HPEN initiation and periodically during treatment using validated questionnaires [5,6]. For patients on HEN, the NutriQol^®^ instrument has been reported as valid and reliable and is considered useful for this patient population [107,108]. Several validated tools have been used to study the quality of life in this patient population including the European Organization for Research and Treatment of Cancer Quality of Life Questionnaire-Core30 (EORTC QLQ-C30), Functional Assessment of Cancer Therapy-General (FACT-G), and Therapy Impact Questionnaire (TIQ) [109,110,111].

Survival of patients requiring HPEN is generally related to the underlying disease process for which HPEN is indicated. The reported 5-year survival rate for patients receiving HPN for non-malignant indications ranges between 58% and 83%, and several studies have shown that most patients who die on HPN do so from complications of their disease rather than as a complication of the PN itself [10,112]. The mortality rate for patients requiring HEN is also variable and, similar to those on HPN, is closely related to the diagnosis and age of the patient [10].

## 9. Conclusions

Home parenteral and enteral nutrition has proven to be a life-saving therapy. Patients receiving HPEN are best served by a multidisciplinary team that is familiar with the various medical conditions, potential complications, and therapeutic options. At least for U.S. consumers, financial considerations and insurance coverage are important factors in the successful provision of HPEN. Patient and caregiver education and routine monitoring are essential to reduce or prevent complications related to HPEN. Involvement with organizations that provide additional resources and support to HPEN patients can enhance their quality of life.

## Figures and Tables

**Table 1 nutrients-14-02558-t001:** Indications for HPEN.

**Selected EN Indications**
• Dysphagia ○ Neurologic disorders such as ALS, systemic sclerosis, Parkinson’s disease, cerebrovascular accident
• Malignancy and/or ongoing treatments such as radiationHypercatabolic states ○ Cystic fibrosis ○ Burns ○ Malignancy ○ Chronic obstructive pulmonary disease ○ Chronic infection
• Preoperative or postoperative malnutrition • Upper gastrointestinal obstruction ○ Esophageal stricture ○ Gastric outlet obstruction (malignancy, pancreatitis, etc.)
• Malabsorptive or maldigestive states ○ Inflammatory bowel disease ○ Exocrine pancreatic insufficiency/chronic pancreatitis ○ Cirrhosis ○ Cystic fibrosis
Severe gastric dysmotility
**Selected PN Indications**
Chronic intestinal obstruction or pseudoobstruction Short bowel syndrome Preoperative or postoperative malnutrition Intestinal injury/trauma High-output stoma or enterocutaneous fistula Inability to supply or maintain nutrition via enteral access

**Table 2 nutrients-14-02558-t002:** Central venous catheters.

Type of Catheter	Duration	Pro	Con
**Peripherally inserted central catheter (PICC)**	Short-term	Ease of insertion and removal Cost-effective Accessibility	May have an increased risk of thrombosis and displacement
**Subcutaneous port**	Long-term	Low risk of infection Easier site care Patient comfort	Requires surgical placement and removal Requires a needle to access the port limiting use in patients who requiring daily line access
**Tunneled catheter**	Long-term	Low risk of infection compared to non-tunneled	Requires surgical insertion
**Non-tunneled catheter**	Short-term	Ease of insertion, can be done at bedside	High rate of infection Patient discomfort

**Table 3 nutrients-14-02558-t003:** General education topics for HPN administration.

Hand hygiene Aseptic technique to access and maintain catheter and catheter site PN administration including multivitamin and insulin additives How to use tubing, caps, and other supplies Starting and stopping PN infusion Programming the infusion pump How and where to obtain supplies How to manage supply or PN contamination Who to contact with questions or concerns

**Table 4 nutrients-14-02558-t004:** Recommended laboratory monitoring for HPN.

Laboratory Timing	Laboratory Studies *
**Baseline**	Complete blood count Comprehensive metabolic panel Prothrombin time/INR Magnesium Phosphorus Zinc Selenium Copper Manganese Vitamin A Vitamin E 25 hydroxyvitamin D Vitamin B12/methylmalonic acid Folate Iron studies Ferritin Parathyroid hormone Essential fatty acids Triglyceride
**Weekly/Biweekly/Monthly** **until stabilization**	Basic metabolic panel Magnesium Phosphorus
**Quarterly after stabilization**	Complete blood count Comprehensive metabolic panel Triglyceride (?)
**Annually**	Same as baseline

* Note: Abnormal vitamin and trace element levels at baseline should be monitored more frequently until levels normalize after supplementation commences.

**Table 5 nutrients-14-02558-t005:** Risk factors for refeeding syndrome.

BMI < 18.5 Unintentional weight loss > 10% of total body weight Little or no nutritional intake Low levels of potassium, magnesium, or phosphate prior to feeding Comorbidities that predispose to malnutrition including anorexia nervosa, malignancy, advanced age, alcohol/substance misuse or abuse

**Table 6 nutrients-14-02558-t006:** Factors affecting the quality of life of patients on HPEN.

Formula infusion time Duration of nutritional support therapies Sleep disturbance related to nutrition infusion and equipment Family and social life disturbance Recreational activity limitations from implanted medical devices Involvement in support group

## Data Availability

Not applicable.

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
