# Peer review of "Home Parenteral and Enteral Nutrition"

_nutrients, 2022, doi:10.3390/nu14132558_

Round 1

Reviewer 1 Report

This manuscript provides a comprehensive review of parenteral and enteral nutrition (HPEN) indications, patient requirements, monitoring, complications, and the overall process of managing these therapies at home. Some minor concerns need to be addressed.

What is more puzzling is why page 27 directly includes the template introduction of MDPI. Careful checking is required in submitting manuscripts.

This manuscript should address how parenteral and enteral nutrition from hospital to home would be covered. Given the fact that parenteral and enteral nutrition are currently used as adjunctive treatment in hospitals, this cannot be ignored and should be discussed with emphasis.

In addition, some guidance adopted by the authors should be noted, such as the adopted guidelines of the European Society of Parenteral Enteralnutrition.

Author Response

This manuscript provides a comprehensive review of parenteral and enteral nutrition (HPEN) indications, patient requirements, monitoring, complications, and the overall process of managing these therapies at home. Some minor concerns need to be addressed.

What is more puzzling is why page 27 directly includes the template introduction of MDPI. Careful checking is required in submitting manuscripts.

Our apologies, we are not sure how this happened. This has been removed from the manuscript.

This manuscript should address how parenteral and enteral nutrition from hospital to home would be covered. Given the fact that parenteral and enteral nutrition are currently used as adjunctive treatment in hospitals, this cannot be ignored and should be discussed with emphasis.

Thank you for this excellent suggestion. A new section, Transitioning from Hospital to Home, has now been added on page 6.

In addition, some guidance adopted by the authors should be noted, such as the adopted guidelines of the European Society of Parenteral Enteralnutrition.

We agree that ESPEN provides excellent guidance regarding both home enteral and parenteral nutrition and several of their guidelines and articles have been referenced throughout this review.

Reviewer 2 Report

\The manuscript "Home Parenteral and Enteral Nutrition" written by Jamie Bering and John DiBaise covers essential issues about managing parenteral and enteral nutrition at home. It is a comprehensive narrative review discussing the indications and requirements, advantages and disadvantages of catheters and enteral access, nutritional formulas, training, and monitoring of enteral and parenteral nutrition. An important part is devoted to the possible complications and, finally, the cost and QoL. This kind of manuscript may be better suitable for a chapter in a textbook but also in a journal as a narrative review. There are some minor issues regarding the editing of Table 3 (probably there were created according to the journal rules, but better not to split some lines). Also, I would improve the abstract as it seems to me too short.      

Author Response

The manuscript "Home Parenteral and Enteral Nutrition" written by Jamie Bering and John DiBaise covers essential issues about managing parenteral and enteral nutrition at home. It is a comprehensive narrative review discussing the indications and requirements, advantages and disadvantages of catheters and enteral access, nutritional formulas, training, and monitoring of enteral and parenteral nutrition. An important part is devoted to the possible complications and, finally, the cost and QoL. This kind of manuscript may be better suitable for a chapter in a textbook but also in a journal as a narrative review. There are some minor issues regarding the editing of Table 3 (probably there were created according to the journal rules, but better not to split some lines). Also, I would improve the abstract as it seems to me too short.      

Thank you for your comments. We agree that Table 3 would be more visually appealing without the line breaks. We suspect this to be a formatting issue that would likely be improved in the final draft.  The abstract has also now been expanded.

Reviewer 3 Report

This is a clearly written, thorough, relevant comprehensive review which achieves its aim of presenting an overview of HPEN, which is incredibly useful to ensure healthcare providers understand key elements of this important therapy, the prevalence of which is increasing globally. This review is well structured, has appropriate tables, highly relevant references and a coherent conclusion, and on publication will be a very useful resource in the field. The main strengths of this review are that it is comprehensive yet concise, and therefore will be a very practical resource for referring to, for up-to-date information on proper management of patients on HPEN. Please see specific comments below.

Line 50-52 states ‘Although there are no formal recommendations regarding training for those receiving HEN, we suggest that education also be provided to reduce the risk of complications at home’.

Can authors expand this to reflect that the Espen guidelines on HEN (Bischoff 2020) advise HEN patients and carers need training by multidisciplinary teams (‘HEN patients and their carers, need training in managing their EN regimens by a multidisciplinary team [78]. Prior to discharge they need to be able to demonstrate competency in feed administration, equipment handling and some basic trouble shooting in case of tube or equipment failure [79].)

Line 57: Can author consider adding reference to Boullata 2017 (Safe practices in Enteral Nutrition) as regards a multi-disciplinary team being needed

Line 109: Suggest edit title to ‘Nutritional Formula and administration’ given the content of the section

Lines 173 – 180 focus solely on PN. Please expand to include brief discussion of training as it relates to the success of HEN also. See Espen guidelines on HEN (2020) Recommendation 50 which discusses this and references key papers.

Line 185 states 'Patients receiving HPN require routine monitoring that should include body weight, urine output, biochemistry lab studies including micronutrient levels, and bone density measurement'. The reference given for this however is the Espen guideline on Enteral (Bischoff 2020). Was it intended to reference Bischoff 2020 here, as opposed to a reference on parenteral nutrition?

Lines 221 – 225 under HEN Monitoring – can authors add a reference for the aspects that should be routinely monitored in HEN, such as Bischoff 2020 – ESPEN recommendation 44 which refers to aspects such as body weight

Line 386-389 The mention of microbial issues although useful, is quite short, and would benefit from being fleshed out moderately to include mention of key issues such as following recommended hang times, and timings around use of administration sets, within the education that patients should receive.

Lines 497 – 511: Would be beneficial to add that supporting patients and caregivers throughout their journey with HEN can help reduce negative impacts these therapies can have on their lives (Day 2017 – Home enteral feeding and its impact on quality of life. Br J Community Nurs, 2017, 22, s14-s16). Mentioning what the state of play is as regards guidelines recommending measurement of QoL routinely in HPEN patients would also be a useful addition to this section.

Other minor issues: Line 495, clarify who should keep these parameters available ; Table 2 - in duration column, remove 'use' after 'long-term', to be consistent with 'short-term'

Author Response

We would like to thank you for your thorough and thoughtful comments and suggestions. These suggested revisions to the manuscript have improved the overall quality of this article.

This is a clearly written, thorough, relevant comprehensive review which achieves its aim of presenting an overview of HPEN, which is incredibly useful to ensure healthcare providers understand key elements of this important therapy, the prevalence of which is increasing globally. This review is well structured, has appropriate tables, highly relevant references and a coherent conclusion, and on publication will be a very useful resource in the field. The main strengths of this review are that it is comprehensive yet concise, and therefore will be a very practical resource for referring to, for up-to-date information on proper management of patients on HPEN. Please see specific comments below.

Line 50-52 states ‘Although there are no formal recommendations regarding training for those receiving HEN, we suggest that education also be provided to reduce the risk of complications at home’.

Can authors expand this to reflect that the Espen guidelines on HEN (Bischoff 2020) advise HEN patients and carers need training by multidisciplinary teams (‘HEN patients and their carers, need training in managing their EN regimens by a multidisciplinary team [78]. Prior to discharge they need to be able to demonstrate competency in feed administration, equipment handling and some basic trouble shooting in case of tube or equipment failure [79].)

Thank you for this feedback. This section has been revised to reflect the noted guidance from ESPEN.

Line 57: Can author consider adding reference to Boullata 2017 (Safe practices in Enteral Nutrition) as regards a multi-disciplinary team being needed

This reference has been added to the manuscript.

Line 109: Suggest edit title to ‘Nutritional Formula and administration’ given the content of the section

Thank you for this suggestion. The title has been updated to “Nutritional Formulation and Administration”

Lines 173 – 180 focus solely on PN. Please expand to include brief discussion of training as it relates to the success of HEN also. See Espen guidelines on HEN (2020) Recommendation 50 which discusses this and references key papers.

This section has been updated and now includes references to studies showing success of HEN training.

Line 185 states 'Patients receiving HPN require routine monitoring that should include body weight, urine output, biochemistry lab studies including micronutrient levels, and bone density measurement'. The reference given for this however is the Espen guideline on Enteral (Bischoff 2020). Was it intended to reference Bischoff 2020 here, as opposed to a reference on parenteral nutrition?

We have corrected this reference which is now updated to the ESPEN guidelines on home parenteral nutrition from 2020 by Pironi et al.

Lines 221 – 225 under HEN Monitoring – can authors add a reference for the aspects that should be routinely monitored in HEN, such as Bischoff 2020 – ESPEN recommendation 44 which refers to aspects such as body weight.

Thank you, this reference has been added to the manuscript.

Line 386-389 The mention of microbial issues although useful, is quite short, and would benefit from being fleshed out moderately to include mention of key issues such as following recommended hang times, and timings around use of administration sets, within the education that patients should receive.

Thank you for this excellent suggestion. A new paragraph has been added on page 10 (lines 411-423) expanding on this important issue.

Lines 497 – 511: Would be beneficial to add that supporting patients and caregivers throughout their journey with HEN can help reduce negative impacts these therapies can have on their lives (Day 2017 – Home enteral feeding and its impact on quality of life. Br J Community Nurs, 2017, 22, s14-s16). Mentioning what the state of play is as regards guidelines recommending measurement of QoL routinely in HPEN patients would also be a useful addition to this section.

This section has been updated to include this reference from Day et al. Another paragraph has been added with a brief review of available literature regarding QoL surveillance in this patient population.

Other minor issues:

Line 495, clarify who should keep these parameters available

This has been updated.

Table 2 - in duration column, remove 'use' after 'long-term', to be consistent with 'short-term'

These changes have been made.